# Risk of Occupational Latent Tuberculosis Infection among Health Personnel Measured by Interferon-Gamma Release Assays in Low Incidence Countries—A Systematic Review and Meta-Analysis

**DOI:** 10.3390/ijerph17020581

**Published:** 2020-01-16

**Authors:** Claudia Peters, Agnessa Kozak, Albert Nienhaus, Anja Schablon

**Affiliations:** 1Competence Center for Epidemiology and Health Services Research for Healthcare Professionals (CVcare), Institute for Health Services Research in Dermatology and Nursing (IVDP), University Medical Centre Hamburg-Eppendorf, 20246 Hamburg, Germany; a.kozak@uke.de (A.K.); albert.nienhaus@bgw-online.de (A.N.); a.schablon@uke.de (A.S.); 2Department of Occupational Medicine, Hazardous Substances and Public Health, Institution for Statutory Accident Insurance and Prevention in the Health and Welfare Services (BGW), 22089 Hamburg, Germany

**Keywords:** latent tuberculosis infection (LTBI), health personnel, occupational risk, interferon-gamma release assays (IGRA), low incidence countries

## Abstract

Healthcare workers (HCWs) have increased risk for latent tuberculosis infection (LTBI) and tuberculosis (TB) disease due to their occupational exposure. For some years now, interferon-γ release assays (IGRAs) have replaced the tuberculin skin test for the diagnosis of LTBI in many countries. This review examined the occupational risk of LTBI in HCWs with IGRA testing in low incidence countries. A systematic review and meta-analysis of studies from 2005 onwards provide data regarding the prevalence of LTBI in HCWs. In addition, the pooled effect estimates were calculated for individual regions and occupational groups. 57 studies with 31,431 HCWs from four regions and a total of 25 countries were analysed. The prevalence of LTBI varied from 0.9 to 85.5%. The pooled estimation found the lowest prevalence of LTBI for North American and West Pacific countries (<5%), and the highest prevalence for Eastern Mediterranean countries (19.4%). An increased risk for LTBI was found only for administrative employees. Studies on the occupational risk of LTBI continue to show increased prevalence of HCWs, even in low-incidence countries. Good quality studies will continue to be needed to describe occupational exposure.

## 1. Introduction

Healthcare workers (HCWs) have an elevated risk of latent tuberculosis infection (LTBI) and TB disease (tuberculosis (TB)) due to the nature of their jobs [1,2,3,4]. The reduction in TB incidence in high-income countries should correlate with a decrease in the risk of TB infection for HCWs. However, it appears that working in the healthcare sector even poses a risk in high-income countries with high hygiene standards [5,6]. In Germany, where TB incidence is low, TB in healthcare workers remains one of the most common infections reported to the compensation board [7].

Most of the reviews of the TB infection risk among HCWs were conducted in low and middle-income countries [3,8,9,10]. Furthermore, the occupational infection risk and the probability of causation have primarily been analysed while using the tuberculin skin test (TST) after Mendel and Mantoux. For several years, interferon-γ release assays (IGRAs) have also been used to diagnose latent tuberculosis infection. IGRAs have higher specificity and a good negative predictive value and are, therefore, a valid alternative to TST. In the review and meta-analysis by Diel et al. [11], the specificity of the IGRAs was found to be 98–100%. The negative predictive value was 97.8% for T-Spot TB and 99.8% for QFT-GIT. Thus, the IGRAs have strong advantage in the diagnosis of LTBI and they can more accurately exclude LTBI. Therefore, they replace the TST in most high income, low incidence countries. 

This systematic review and meta-analysis examines the prevalence and occupational risk of LTBI assessed by IGRA in healthcare workers in low-incidence countries.

## 2. Methods

This literature review was conducted in accordance with the Preferred Reporting Items for Systematic Reviews and Meta-Analyses (PRISMA) [12]. The Meta-analysis of Observational Studies in Epidemiology (MOOSE) reporting checklist was also taken into account [13].

The question and the corresponding inclusion and exclusion criteria were formulated while using the PEO criteria:

**P**opulation: Healthcare workers from countries with low tuberculosis (TB) incidence

**E**xposure: Occupational exposure to TB pathogens, infected material, or an infected environment

**O**utcome: LTBI found in the occupational setting using IGRAs

(1)How high is the prevalence of LTBI among healthcare workers in low-incidence countries, measured using IGRAs?(2)In which occupational groups or areas of work within the healthcare sectors is there an elevated risk of occupational LTBI?

### 2.1. Selection Criteria

The target population was defined as healthcare workers whose occupations meant that they had either direct contact with patients (doctors, nursing staff and assistants, students, various therapists) or indirect or no contact with patients, but were exposed to infected material or an infected environment (e.g., laboratory workers, cleaning staff, administrative employees). The review only included studies that were conducted in countries with a low incidence of TB, as defined by the WHO (estimated incidence rate <40 per 100,000 inhabitants [14]). Studies from low-incidence countries with a study population exclusively from high-incidence countries were excluded due to the selective group. Furthermore, the occupational exposure to LTBI had to have been investigated while using immuno-diagnostic tests as part of routine examinations or screenings. Studies that used both IGRAs and TSTs as diagnostic procedures were only included if the results of these two methods were recorded separately. Cases where IGRAs were used to confirm a positive TST were excluded from further analysis on the grounds of selection. Studies with contact investigations following the disclosure of active TB cases were likewise excluded. 

Observational studies comprised cohort studies, case control studies, and cross-sectional studies. Reviews, editorials, comments, conference reports, case reports, and statements were excluded. Abstracts were only taken into account if they included all of the relevant information and the full text was not available. The initial study had no language restrictions. However, at the full text screening stage, only peer-reviewed publications in a language spoken by the study group were included (e.g., English, German, Dutch, French, Italian, Portuguese, and Spanish). A detailed overview of the inclusion and exclusion criteria can be found in Appendix A.

### 2.2. Sources of Information and Search Strategy

A systematic electronic search was conducted in the MEDLINE, PubMed, CINAHL, Web of Science and LIVIVO databases. The search string was initially devised for PubMed and then adapted to other databases. The keywords were developed in accordance with Medical Subject Headings (MeSH) and then combined with additional terms for the target population, exposure/diagnostic procedure, and the outcome. The detailed search strategy and database search in PubMed can be found in Appendix A. The search spanned the publication period from 1 January 2005 to 31 January 2019 (last update: 15 August 2019).

### 2.3. Data Management, Study Selection and Data Extraction

The search results were sorted and managed while using the referencing software EndNote. First, the citations were automatically checked for duplicates; then, the search results were manually cleaned up. The titles and abstracts were selected by a reviewer (AK) using the predefined inclusion and exclusion criteria. Two reviewers checked the full text of the shortlisted articles for relevance (AK and CP). In the case of disagreements, an additional reviewer was consulted (AS) to reach a consensus through discussion. A manual search was conducted for other relevant sources in the references of the identified publications. Systematic reviews and meta-analyses were also included. One reviewer (CP) extracted the data, then compared and verified by a second reviewer (AK). The following information was extracted from all of the studies using a prescribed form: lead author, publication date, country, study design, study period, research setting, occupational group, age, diagnostic method, IGRA cut-off values, other risk factors for LTBI, and proportion of positive IGRA results. Finally, the studies were listed by WHO region.

### 2.4. Study Quality

The studies’ methodological quality was assessed using the JBI Critical Appraisal Checklist for Analytical Cross Sectional Studies [15]. This checklist consists of eight items which address biases in the design, execution, and analysis of the studies. One item (“Were objective, standardised criteria used to measure the findings?”) was replaced with another item from the JBI Checklist for Studies Reporting Prevalence Data (“Was the sample size adequate?”). While using the sample calculation formula by Naing et al. [16], which is cited in the description, a sample size of N = 139 was calculated for an anticipated LTBI prevalence of 10%. Consequently, in the quality appraisal, a point was awarded if at least this number of study participants was included in individual studies. The maximum score was eight points, one point for each question. High, medium, and low-quality studies were defined as scoring 7–8, 5–6, and less than 5 points, respectively. Two reviewers independently evaluated the study quality (CP, AS).

1st item:  Criteria for inclusion in the sample clearly defined2nd item: Study subjects and setting described in detail3rd item: Exposure measured in a valid and reliable way4th item: Sample size adequate (min. N = 139)5th item: Confounding factors identified/considered6th item: Strategies for dealing with confounding factors indicated7th item: Outcomes measured in a valid and reliable way8th item: Appropriate statistical analysis applied

### 2.5. Statistical Analysis

For the LTBI prevalence, the number of positive LTBI cases and total number of participating healthcare workers was extracted from the original studies into an Excel table that was developed by Neyeloff et al. [17]. On this basis, pooled prevalence estimates were calculated with a 95% confidence interval (CI). Random effects models were chosen to determine the pooled prevalence estimates for the individual WHO regions, as we assumed that the effects between studies were heterogeneous (European, Western Pacific, Eastern Mediterranean, and Americas). 

The corresponding number of positive LTBI cases in the respective occupational group/field of work (e.g., nursing staff, doctors, administrative employees or laboratory workers) was compared with the case count in the whole study population. Only studies with good or very good methodological scores (≥5 points) were included in the meta-analysis. The Mantel-Haenszel method was used to calculate odds ratios (OR) for dichotomous outcomes. The model calculations are based on random effects. Forest plots were generated for each group comparison. Heterogeneity was quantified with the aid of the chi-square (χ^2^) and I^2^ statistic. The latter expresses the total variability between the studies as a percentage. The higher the percentage, the greater the degree of heterogeneity: I^2^ values between 0–40%, 30–60%, 50–90%, and 75–100% correspond to a low, medium, substantial, or high level of heterogeneity A *p*-value (χ^2^) of <0.10 was deemed to be statistically significant. Review Manager (version 5.3), which was the statistical software provided by Cochrane, was used for data analysis [18].

## 3. Results

### 3.1. Study Selection

1308 matches were identified in the databases and from other sources (Figure 1). 254 full texts were screened after removing the duplicates and screening titles and abstracts. The most common reasons for exclusion were: publications covered contact investigations following TB exposure; there was no medical study population or the study took place in high-incidence countries; there was no use of IGRA tests or its use was only to confirm a positive TST; or, the exclusion was due to the type of publication. Appendix A lists the exact selection criteria.

A total of 57 studies fulfilled the inclusion criteria and could be included in the analysis. Of these, 32 studies were conducted in Europe, five in America, nine in the Western Pacific region, and 11 in countries in the Mediterranean region (Table 1). The countries were allocated according to WHO regions.

### 3.2. Study Characteristics

Most of the studies were conducted as cross-sectional studies and cohort studies in hospital settings between 2005 and 2015. Only staff from special infection or TB units were included in some of the investigations. Less commonly, the studies looked at university settings, laboratories, radiology departments, or geriatric care. The number of participants varied between 21 and 3823 employees. For the most part, the IGRA testing for LTBI used the QFT (various generations) with a cut-off value for INF-γ of 0.35 IU/mL. The T-SPOT assay was used six times and a non-commercial ELISpot was utilised, alongside the other two tests in the study by Girardi et al. [23]. An appraisal of the studies’ methodological quality resulted in a high score for 31 studies, while 15 studies were awarded a medium score. 10 studies were deemed to be of low methodological quality. For one study [26], only an abstract was available, so it could not be included in the evaluation. The most common shortcomings were insufficient study size, the approach for dealing with confounding factors and a lack of suitable statistical methods for the identification of risk factors (e.g., regression analysis).

### 3.3. Meta-Analysis—Prevalence of LTBI

The prevalence of LTBI in the individual studies was between 0.9% and 85.5%. The findings from Europe varied particularly widely, from 1.1 to 85.5%. 

The pooled effect estimates for Europe showed an LTBI prevalence of 16.2% (95% CI 13.0–19.3) (Table 2). When all of the studies with high and medium methodological quality were considered, the estimate was 16.3%. The figure was 13.9% among those with at least 139 employee participants. For America, the joint effect estimate indicated a prevalence of 16.5%, while excluding low quality studies gave a prevalence of 19.3%. Excluding one study with a small number of participants reduced the estimate to 14.9% [54]. However, the prevalence was 4.5% if only North America is considered. The pooled prevalence for Western Pacific countries was 4.8%, while 19.4% was recorded for the Eastern Mediterranean region. Selection on the basis of high and medium study quality reduced the joint prevalence in the Mediterranean region to 16.1%; looking only at those with a sufficient cohort size resulted in 15.2%. Seven of the nine studies in the Western Pacific region were conducted in Japan. No studies were excluded here, so there was no variation in the findings.

### 3.4. Meta-Analysis—Occupational Risk for LTBI

23 studies were available for the meta-analysis to establish occupational exposure. Studies whose methodological quality was considered low were excluded from the analysis [29,47], as was another study, which focused solely on laboratory staff [73]. All in all, 20 studies with data from 2612 cases of LTBI among a total population of 15,262 workers were included in the meta-analysis (Table 3). The majority of the publications came from the Eastern Mediterranean and Western Pacific regions. It was not possible to extract any data to analyse the occupational exposure from the studies that were conducted in America. 

Stratification by occupation revealed a statistically significant exposure risk for administrative employees, with an OR of 1.6 (95% CI 1.18–2.17) (Figure 2). Forest plots for other occupations can be found in Appendix A. The studies from Europe confirm this finding (OR 1.7). In the other regions, only a few studies showed an elevated, but not statistically significant OR. An elevated risk (OR 2.6) was identified for laboratory staff in the Western Pacific region, which was not evident in the other regions. Doctors and nurses were not found to be at any greater risk of exposure than other workers. By contrast, a statistically insignificant protective effect was identified among the nursing staff. Even after regional differentiation, this was still observed for Europe and the Western Pacific region. The heterogeneity of the meta-analyses can be largely seen as moderate.

## 4. Discussion

57 studies investigated the occupational LTBI risk of medical personnel in low-incidence countries using the IGRA tests. The prevalence varied between 0.9% and 85.5%. Regional stratification revealed the lowest pooled effect estimates for North American and Western Pacific countries (<5%) and the highest for the Eastern Mediterranean (19.4%). The job-related analysis showed an increased infection risk for administrative staff, whereas there was no indication of an elevated risk for doctors and nurses in these studies. 

In the review by Apriani et al. [9], the highest prevalence of 60% for LTBI that was measured with IGRA tests was identified among general service staff, followed by doctors (35%) and nurses (34%). General service staff included cleaners, drivers, and housekeepers. However, this review targeted low and middle-income countries; high-incidence countries were not an exclusion criterion. In a meta-analysis, Uden et al. [8] investigated the LTBI prevalence among medical staff with a comparison group consisting of employees with no direct patient contact or the general population. This found an elevated risk of LTBI and a higher incidence of active TB among healthcare workers. However, there was no stratification for the individual occupational groups. The pooled prevalence estimate was 37% and the risk estimate put the OR at 2.3 (95% CI 1.6–3.21) for LTBI. However, the TST and IGRA results were analysed together and no constraints were applied to the TB incidence. Baussano et al. [76] also identified an elevated risk for healthcare workers as compared with the general population in low-incidence countries. In a further review, the LTBI risk in high-burden countries was investigated using TSTs. This found a pooled LTBI prevalence of 57% for medical personnel [10].

Many of the studies analysed individual risk factors alongside LTBI prevalence. This resulted in 16 studies identifying higher age as a risk factor, while eight studies found staff from high-incidence countries to be at greater risk. In an occupational context, length of service, workplaces in units with high exposure, and specific job categories were repeatedly listed. The various studies’ findings differ with regard to job category. For instance, Torres Costa [49] and Bukhary et al. [68] found that doctors had an elevated risk vis-à-vis administrative staff. Two further studies identified an elevated risk for doctors and nurses [64,77]. However, another study showed elevated risk for administrative staff [36]. Our meta-analysis found no risks for medical professionals such as doctors, nurses and laboratory staff, but it did show an occupational risk of LTBI for administrative employees. However, most of the literature does not exactly define who was included in this group. It could encompass receptionists, administrative staff, or service and managerial employees, for example. This makes it difficult to venture an explanation for this finding.

In this review, studies of LTBI prevalence in low-incidence countries using the IGRA diagnostic procedure were collectively viewed. Regions and occupational groups were also analysed. Seidler et al. [5] previously formulated the same study objective in relation to TSTs. The authors found an elevated risk for various medical professions and/or units, although they describe the epidemiological evidence for the occupational groups as limited, except in the case of nurses. One reason was the lack of methodologically suitable studies with sufficient power.

Numerous studies were available for this review based on the use of IGRA testing among healthcare workers. Combining the studies offered greater statistical power and more meaningful findings than was possible with individual pieces of research. However, the result of the joint effect estimate can only be interpreted in relation to the underlying data of the individual publications. Conducting a quality appraisal of the studies, completing sub-group analyses and using the random effects model addressed the problem of statistical heterogeneity. Other criteria—such as the known risk factors of age or origins in a high-incidence country—could not be taken into account when calculating the pooled estimates due to the heterogeneity of the data. Most of the studies used convenience samples, which were tested with IGRA, due to some kind of risk assessment. Therefore, it is difficult to define an unexposed group, rending risk analysis unreliable.

Our strict inclusion criteria regarding the consideration of age or occupation in the individual studies must be viewed as a limitation. Nevertheless, although many of the included studies investigated these factors, they did not publish data on them. For this reason, the large studies from North America [51,52,53], for example, could be included in the overview, but not in the meta-analysis. A further limitation is the non-consideration of the results of the repeated test with the IGRA. For serial testing studies, we have only included the baseline test results for analysis in this review to ensure comparability with studies that only involved a single IGRA test. Repeated testing can lead to reversions and thus, if not taken into account, can lead to a possible overestimation of the risk of LTBI in HCWs.

## 5. Conclusions

This systematic review presents studies on the occupational LTBI risk of healthcare workers in low-incidence countries, as measured using the IGRA diagnostic procedure. It became apparent that numerous studies had been conducted on this topic and that prevalence is widespread. Regional differentiation showed the lowest LTBI prevalence for the North American and Western Pacific region and the highest for Eastern Mediterranean countries. A breakdown by occupational group identified an elevated infection risk for administrative employees, which was unexpected. Well-designed cohort studies are warranted for describing the occupational risk for LTBI in healthcare workers in low incidence countries.

## Figures and Tables

**Figure 1 ijerph-17-00581-f001:**
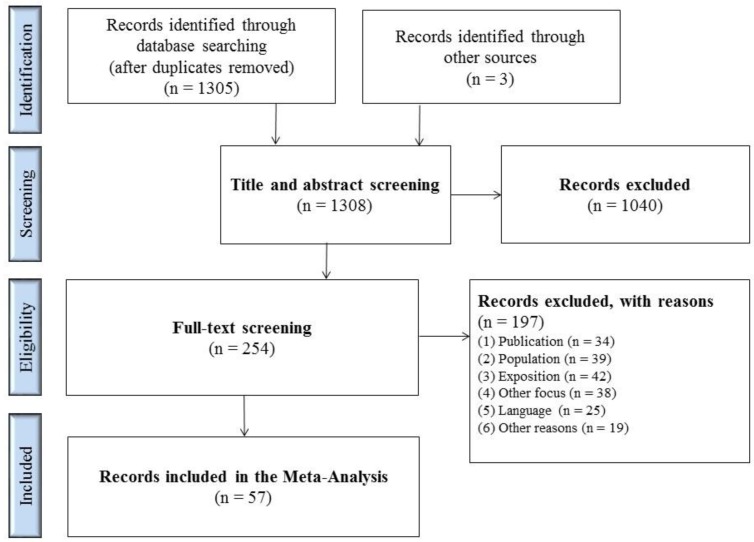
Flowchart of study selection process.

**Figure 2 ijerph-17-00581-f002:**
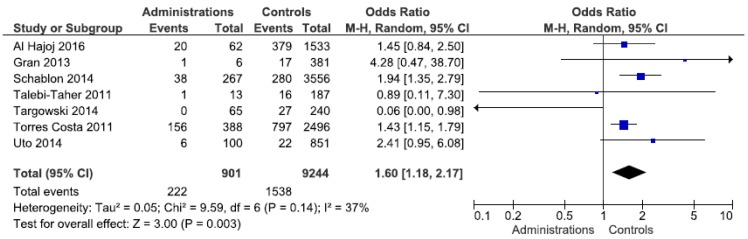
Forest plot of the LTBI prevalence in administrative employees by IGRA in low incidence countries.

**Table 1 ijerph-17-00581-t001:** Studies of occupational latent tuberculosis infection (LTBI) with interferon-γ release assays (IGRAs) by WHO regions

1st Author, yr	Country	Study Design	Study Period	Setting	IGRA Used	No. of HCWs	No. of LTBI	Prevalence (%)	95% CI	Quality
**Europe**										
Fox 2009 [19]	Israel	cross-sectional	2007	outpatient TB center	QFT-GIT	100	17	17.0	10.8–25.7	++
Soborg 2007 [20]	Denmark	cross-sectional	2007	infectious disease ward	QFT-GIT	139	2	1.4	0.7–5.4	+++
Gran 2013 [21]	Norway	cross-sectional	2008–09	hospital/TB ward	QFT-GIT	387	18	4.7	2.9–7.3	+++
Ciaschetti 2007 [22]	Italy	cross-sectional	2006–07	hospital	QFT-GIT	590	63	10.7	8.4–13.4	+++
Girardi 2009 [23]	Italy	cross-sectional	2004–05	hospital/TB ward	QFT-GIT T-SPOT.TB ELISPOT	115 115 115	29 42 40	25.2 36.5 34.8	18.1–33.9 28.3–45.6 26.7–43.9	+++
Larcher 2012 [24]	Italy	cross-sectional	2006–07	hospital	QFT-GIT	621	115	18.5	15.7–21.8	+++
Sauzullo 2014 [25]	Italy	cross-sectional	n/a	hospital	QFT-GIT	196	34	17.3	12.7–23.3	++
Magrini 2016 [26]	Italy	cross-sectional	2007–13	hospital	QFT-GIT	939	46	4.9	3.7–6.5	n.a.
Stebler 2008 [27]	Switzerland	retrospective	2005–06	hospital	QFT-GIT	777	59	7.6	5.9–9.7	+++
Tripodi 2009 [28]	France	cross-sectional	2006–07	hospital	QFT-GIT	148	28	18.9	13.4–26.0	+++
Faibis 2011 [29]	France	cross-sectional	2008	hospital	QFT-GIT	137	16	11.7	7.2–18.2	+
Moucaut 2013 [30]	France	cross-sectional	2007–11	hospital	QFT-GIT	634	141	22.2	19.2–25.6	+++
Nienhaus 2014 [31]	France	prospective	2008–13	hospital	QFT-GIT	1192	265	22.2	20.0–24.7	+++
Lucet 2015 [32]	France	prospective	2009–10	hospital	QFT-GIT	807	113	14.0	11.8–16.6	++
Barsegian 2008 [33]	Germany	cross-sectional	2006	radiology department	T-SPOT.TB	95	1	1.1	<0.01–6.3	+
Schablon 2009 [34]	Germany	cross-sectional	2005–08	TB hospital	QFT-GIT	265	19	7.2	4.6–11.0	+++
Schablon 2011 [35]	Germany	prospective	2008–11	nursing school	QFT-GIT	194	4	2.1	0.6–5.4	++
Schablon 2014 [36]	Germany	prospective	2006–13	hospital + nursing home	QFT-GIT	3823	318	8.3	7.5–9.2	+++
Herzmann 2017 [37]	Germany	prospective	2008–14	respiratory hospital	QFT-GIT/ T-SPOT.TB	280	109	38.9	33.4–44.8	++
Khanna 2009 [38]	United Kingdom	cross-sectional	n/a	hospital	QFT-GIT	171	13	7.6	4.4–12.7	++
Alvarez-Leon 2009 [39]	Spain	cross-sectional	2007	hospital	QFT-GIT	134	8	5.97	2.9–11.5	+++
Casas 2009 [40]	Spain	cross-sectional	2004–05	hospital	QFT-GIT T-SPOT.TB	147 147	43 57	29.3 38.8	22.5–37.1 31.3–46.9	+++
Martinez-Lacasa 2015 [41]	Spain	cross-sectional	2010–11	hospital	QFT-GIT	226	17	7.5	4.7–11.8	++
Topic 2009 [42]	Croatia	cross-sectional	2007	hospital	QFT-GIT	54	17	31.5	20.6–44.8	+
Targowski 2014 [43]	Poland	cross-sectional	n/a	hospital	QFT-GIT	305	27	8.9	6.1–12.6	+++
Ozdemir 2007 [44]	Turkey	cross-sectional	2005	hospital	QFT-GIT	76	65	85.5	78.7–71.9	++
Caglayan 2011 [45]	Turkey	cross-sectional	2005	TB hospital	QFT-GIT	78	34	43.6	33.1–54.6	+
Babayigit 2014 [46]	Turkey	cross-sectional	n/a	hospital	QFT-GIT	96	19	19.8	13.0–28.9	+++
Bozkanat 2016 [47]	Turkey	cross-sectional	2008	TB hospital	QFT-GIT	34	7	20.6	10.1–37.1	+
Kargi 2017 [48]	Turkey	cross-sectional	n/a	hospital	QFT-GIT	100	23	23.0	15.8–32.2	+++
Torres Costa 2011 [49]	Portugal	prospective	2007–10	hospital	QFT-GIT	2884	953	33.0	31.4–34.8	+++
Nikolova 2013 [50]	Bulgaria	cross-sectional	2009	TB hospital	QFT-GIT	21	10	47.6	28.3–72.7	+
**The Americas**										
Joshi 2012 [51]	USA	retrospective	2008–09	hospital	QFT-GIT	3290	129	3.9	3.3–4.6	+
Dorman 2014 [52]	USA	cross-sectional	2008–11	hospital	QFT-GIT T-SPOT.TB	2418 2418	118 144	4.9 6.0	4.1–5.8 5.1–7-0	++
Zwerling 2012 [53]	Canada	cross-sectional	2007–11	hospital	QFT-GIT	388	24	6.2	4.2–9.1	+++
Hernandez 2014 [54]	Chile	cross-sectional	2010–11	hospital	QFT-GIT	76	20	26.3	17.7–37.2	+
Ochoa 2017 [55]	Colombia	cross-sectional	2013–15	hospital	QFT-GIT	988	466	47.2	44.1–50.3	++
**Western Pacific**										
Vinton 2009 [56]	Australia	cross-sectional	n/a	hospital	QFT-GIT	481	32	6.7	4.7–9.3	+++
Freeman 2012 [57]	New Zealand	cross-sectional	2007–08	hospital	QFT-GIT	325	28	8.6	6.0–12.2	+++
Harada 2006 [58]	Japan	cross-sectional	2003	hospital	QFT-GIT	332	33	9.9	7.7–13.7	+++
Hotta 2007 [59]	Japan	cross-sectional	2006	hospital	QFT-2G	207	3	1.4	0.3–4.4	++
Adachi 2013 [60]	Japan	cross-sectional	2011–12	hospital	QFT-GIT	165	18	10.9	6.9–16.7	+++
Ogiwara 2013 [61]	Japan	retrospective	2010–11	hospital	QFT-GIT	585	5	0.9	0.3–2.1	++
Uto 2014 [62]	Japan	prospective	2007–10	hospital	QFT-2G	951	28	2.9	2.0–4.2	+++
Mukai 2017 [63]	Japan	cross-sectional	2008–11 2011–14	hospital	QFT-GIT T-SPOT.TB	140 140	6 8	4.3 5.7	1.8–9.2 2.8–11.0	+++
Tanabe 2017 [64]	Japan	cross-sectional	2015	hospital	QFT-GIT T-SPOT.TB	654	19 28	2.9 4.3	1.8–4.5 3.0–6.1	+++
**Eastern Mediterranean**										
El-Helaly 2014 [65]	Saudi Arabia	cross-sectional	2009–11	hospital	QFT-GIT	1412	333	23.6	21.4–25.9	+++
Hassan 2014 [66]	Saudi Arabia	cross-sectional	2012	laboratory	QFT-GIT	134	26	19.4	13.6–27,0	+++
Al Hajoj 2016 [67]	Saudi Arabia	cross-sectional	2012–15	hospital	QFT-GIT	1595	399	25.0	23.0–27.2	+++
Bukhary 2018 [68]	Saudi Arabia	cross-sectional	2015	hospital	QFT-GIT	520	56	10.8	8.4–13.7	++
El-Sokkary 2015 [69]	Egypt	cross-sectional	2012–13	chest hospital/nephrology ward	QFT-GIT	132	38	28.8	21.7–37.1	+++
Hefzy 2016 [70]	Egypt	cross-sectional	2015–16	hospital	QFT-GIT	39	4	10.3	3.5–24.2	++
Talebi-Taher 2011 [71]	Iran	cross-sectional	2009–10	hospital	QFT-GIT	200	17	8.5	5.3–13.3	+++
Salmanzadeh 2016 [72]	Iran	cross-sectional	n/a	hospital	QFT-GIT	87	27	31.0	22.3–41.4	+
Mostafavi 2016 [73]	Iran	cross-sectional	2013–14	laboratory	QFT-GIT	244	42	17.2	13.0–22.5	+++
Keshavarz Valian 2019 [74]	Iran	cross-sectional	2016	hospital	QFT-GIT	101	47	46,5	37.1-56.2	+
Guanche Garcell 2014 [75]	Qatar (Cuban staff)	cross-sectional	2012–13	hospital	QFT-GIT	202	6	3.0	1.2–6.5	++

n/a—not available; Study quality: +++ high, ++ moderate, and + low quality.

**Table 2 ijerph-17-00581-t002:** Pooled prevalence estimations for LTBI by WHO regions.

	Studies (n)	Prevalence (%)	95% CI
**Europe**			
All studies	32	16.4	13.1–19.6
Study quality (+++/++)	25	16.3	12.6–20.1
≥139 participants	20	13.9	10.1–17.7
**The Americas**			
All studies	5	16.5	9.8–23.2
Study quality (+++/++)	3	19.3	1.6–36.9
≥139 participants	4	14.9	7.8–22.0
North America	3	4.5	3.8–5.2
**Western Pacific**			
All studies	9	4.8	3.0–6.6
Study quality (+++/++)	9	4.8	3.0–6.6
≥139 participants	9	4.8	3.0–6.6
**Eastern Mediterranean**			
All studies	11	19.4	13.0–25.7
Study quality (+++/++)	9	16.1	9.5–22.7
≥139 participants	7	15.2	7.9–22.6

Study quality: +++ high, ++ moderate.

**Table 3 ijerph-17-00581-t003:** Meta-analysis for job categories by WHO regions.

WHO Region	Studies (n)	Job Category (n Events/n Total)	Total HCWs (n)	OR	95% CI	I² (%)
		**Nurses**				
**All regions**	20	1072/7077	15,262	0.87	0.72–1.05	59
**Europe**	10	608/4106	9099	0.80	0.63–1.02	56
**Western Pacific**	4	52/1066	2102	0.85	0.58–1.26	0
**Eastern Mediterranean**	6	412/1905	4061	1.01	0.74–1.38	53
		**Physicians**				
**All regions**	19	442/2492	15,130	1.01	0.76–1.36	67
**Europe**	10	354/1616	9099	1.06	0.77–1.45	56
**Western Pacific**	4	27/439	2102	1.09	0.50–2.39	50
**Eastern Mediterranean**	5	61/437	3929	0.92	0.39–2.18	83
		**Lab workers**				
**All regions**	7	24/235	1732	1.01	0.58–1.77	28
**Europe**	3	9/136	583	0.67	0.29–1.56	12
**Western Pacific**	2	7/33	497	2.57	1.05–6.29	0
**Eastern Mediterranean**	2	8/66	652	0.74	0.34–1.65	0
		**Administrations**				
**All regions**	7	222/901	10,145	1.60	1.18–2.17	37
**Europe**	4	195/696	7369	1.69	1.06–2.67	58
**Western Pacific**	1	6 /100	951	2.41	0.95–6.08	na
**Eastern Mediterranean**	2	21 /75	1795	1.41	0.83–2.38	0

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
