# Peer review of "Risk of Occupational Latent Tuberculosis Infection among Health Personnel Measured by Interferon-Gamma Release Assays in Low Incidence Countries—A Systematic Review and Meta-Analysis"

_ijerph, 2020, doi:10.3390/ijerph17020581_

Round 1
Reviewer 1 Report
Overall, I say "Bravo!" to the authors for collating this data. I have some relatively minor observations/suggestions:
1. In line 136, I wonder why a p value of 0.10 was chosen over 0.05. This choice might allow more probability for a determination of significance that might otherwise be due to random chance.
2. In Table 2, the word "Amerika" is chosen versus "America" in the text. Please choose one, depending on your anticipated audience.
3. There were only 5 studies selected in the Americas (as we like to make the distinction), and there is a clear distinction between the three studies from North America and those two from South America. I suspect that if they were separated, the North American statistics would rival those of eastern Asia. Of note, CDC recommendations in the USA have recently eliminated the annual testing in moderate-risk facilities, largely based on the premise that the prevalence of LTBI among HCWs is actually lower than in the general population.
4. The prevalence of LTBI based on IGRA may be spuriously high depending on how a "positive" test is defined, particularly if it entails ANY positive test in a series of tests for an individual. With a cut-off of 0.35 IU/ml, there is a relatively high false-positive rate for QFT, the predominant tool in this study, particularly in workers with a previous history of negative TST. Studies have shown that if an individual has an initial QFT between 0.35 and 1.1, subsequent testing (even within days) will result in a reversion to a value of less than 0.35 in 76% of cases. Thus, without a consistent re-testing policy, reported rates of LTBI based on a single IGRA test are likely spuriously high. This study does not identify those studies which included a retesting policy.
This issue could be addressed in the discussion.
Otherwise, this paper deserves publication.
Thanks for the opportunity to review it.
Author Response
Response to Reviewer 1 Comments
Thanks for the helpful comments to improve the manuscript!
In line 136, I wonder why a p value of 0.10 was chosen over 0.05. This choice might allow more probability for a determination of significance that might otherwise be due to random chance.We made this choice on the basis of the Cochran Handbook, where attention was drawn to the handling of heterogeneity and its problems.
“…Care must be taken in the interpretation of the chi-squared test, since it has low power in the (common) situation of a meta-analysis when studies have small sample size or are few in number. This means that while a statistically significant result may indicate a problem with heterogeneity, a non-significant result must not be taken as evidence of no heterogeneity. This is also why a P value of 0.10, rather than the conventional level of 0.05, is sometimes used to determine statistical significance …” (Cochrane Handbook).
In Table 2, the word "Amerika" is chosen versus "America" in the text. Please choose one, depending on your anticipated audience.
Thank you for the good advice. We have changed this accordingly.
There were only 5 studies selected in the Americas (as we like to make the distinction), and there is a clear distinction between the three studies from North America and those two from South America. I suspect that if they were separated, the North American statistics would rival those of eastern Asia. Of note, CDC recommendations in the USA have recently eliminated the annual testing in moderate-risk facilities, largely based on the premise that the prevalence of LTBI among HCWs is actually lower than in the general population.
Many thanks for the indication. We have now presented the results for North America separately in the manuscript.
The prevalence of LTBI based on IGRA may be spuriously high depending on how a "positive" test is defined, particularly if it entails ANY positive test in a series of tests for an individual. With a cut-off of 0.35 IU/ml, there is a relatively high false-positive rate for QFT, the predominant tool in this study, particularly in workers with a previous history of negative TST. Studies have shown that if an individual has an initial QFT between 0.35 and 1.1, subsequent testing (even within days) will result in a reversion to a value of less than 0.35 in 76% of cases. Thus, without a consistent re-testing policy, reported rates of LTBI based on a single IGRA test are likely spuriously high. This study does not identify those studies which included a retesting policy. This issue could be addressed in the discussion.
It is of course true that single testing may lead to overestimation, as serial testing studies in HCWs have shown that reversions are not rare. However, many studies did not repeat testing, and if they did, we only included the initial testing for comparison. The studies on serial testing often have very different time intervals and are therefore heterogeneous in addition to other factors. We have included this aspect under the limitations as follows:
A further limitation is the non-consideration of the results of the repeated test with the IGRA. For serial testing studies, we have included only the baseline test results for analysis in this review to ensure comparability with studies that have used only a single IGRA test. Repeated testing can lead to reversions and thus, if not taken into account, can lead to a possible overestimation of the risk of LTBI in HCWs.
Reviewer 2 Report
In this review, the authors did a comprehensive literature search. They studied the risk of latent tuberculosis infection and TB amongst the health care workers in countries with a low prevalence rate, diagnosed by the IGRA test. The analysis showed that the rate of LTBI prevalence is high amongst the administrative staff as compared to doctors and nurses.
Minor comments:
page 2, line 67, and page 1, line 42, authors are using the same abbreviation—IGRAs for immunodiagnostic test and interferon-gamma release assays. It is confusing for the reader. Table 3, what is total (n)? For tables, wherever possible while presenting the data, authors should use circular graphs for showing % using different colors. It will be easy to understand. Authors should improve the introduction section and cite the following reference: European Respiratory Journal 2011 37: 88-99; DOI: 10.1183/09031936.00115110
Author Response
Response to Reviewer 2 Comments
Thank you for the helpful comments to improve the manuscript!
Minor comments:
page 2, line 67, and page 1, line 42, authors are using the same abbreviation—IGRAs for immunodiagnostic test and interferon-gamma release assays. It is confusing for the reader.
Thank you for the advice, we have corrected the mistake.
Table 3, what is total (n)?
Total means the total number of all HCWs examined. We have adjusted the descriptions in the table.
For tables, wherever possible while presenting the data, authors should use circular graphs for showing % using different colors. It will be easy to understand.
Your point is well taken. However, we have a lot of information, so we think that tables are suitable for this work. . Besides, colourful graphics in journals are rather the exception.
Authors should improve the introduction section and cite the following reference: European Respiratory Journal 2011 37: 88-99; DOI: 10.1183/09031936.00115110
We have revised the introduction and included the recommended source as follows:
IGRAs have a higher specificity and a good negative predictive value and are therefore a valid alternative to TST. In the review and meta-analysis by Diel et al. the specificity of the IGRAs was found to be 98-100%. The negative predictive value was 97.8% for T-Spot TB and 99.8% for QFT-GIT. Thus, the IGRAs have a strong advantage in the diagnosis of LTBI and can more accurately exclude LTBI.